# Thermoplastic Mandrel for Manufacturing Composite Components with Complex Structure

**Xishuang Jing [1,2], Siyu Chen [1,2], Jiuzhi An [1], Chengyang Zhang [1,3,\*] and Fubao Xie [2]**

[1] School of Mechanical Engineering and Automation, Beihang University, Beijing 100191, China; tom@buaa.edu.cn (X.J.); Xavier_csy@buaa.edu.cn (S.C.); SY2007312@buaa.edu.cn (J.A.)
[2] Beihang Hangzhou Innovation Institute Yuhang, Hangzhou 311121, China; xiefb@buaa.edu.cn
[3] Key Laboratory of Intelligent Manufacturing Technology for Aeronautics Advanced Equipment, Ministry of Industry and Information Technology, Beihang University, Beijing 100191, China
\* Correspondence: zhangchengyang@buaa.edu.cn

**Abstract:** This study was to solve the mandrel demolding problem after curing the composite component with complex structure. In this paper, a reusable thermoplastic mandrel with heating softening characteristics was developed by resin transfer molding (RTM). The glass transition temperature (Tg), surface roughness, and reusability of the mandrel, as well as the shape, surface roughness, thickness uniformity, and internal quality of the formed structure, were tested. The result showed that the Tg of the mandrel was between 80 and 90 °C and the surface roughness was less than Ra 0.5 μm. Additionally, the mandrel can be recycled and can still maintain a good shape after 20 times of deformation. By using this method, the demolding process can be realized by heating and softening the mandrel. The profile error of the formed structure was within 0.5 mm, the surface roughness was less than Ra 0.5 μm, the thickness error was within 0.2 mm, and the average porosity of the upper and lower halves of composite parts was 0.72% and 0.61%. All those data showed that the formed part was in good shape and of good quality. The thermoplastic mandrel can solve the demolding problem of composite materials with complex shapes.

**Keywords:** composites; composites' manufacturing; reusable thermoplastic mandrel; mandrel demolding





## 1. Introduction

Carbon fiber composite was developed rapidly in recent decades. Owing to its light weight, high specific strength, and specific modulus, it can reduce the weight of an aircraft. Many composite components are now made of carbon fiber instead of aluminum alloy [1–3]. With the rapid expansion of its application, carbon fiber composite was gradually used to manufacture components with complex shapes, such as wing spar beams with double torsion, S-shaped variable cross-section inlet, and other parts [4,5]. However, since the forming and curing processes of the aforementioned components are completed at the same time, when the components have cavities, we need to use the mandrel for composite winding [6–8] or prepreg paving to form such a complex structure [9–12]. When we use the traditional hard-mandrel to make a complex structure, after curing the prepreg, the state of the mandrel changes from a viscoelastic state to a solid state, which makes it difficult to demold and even impossible to make a complex structure or composite component with inner cavities.

In order to solve the problem of difficult demolding, experts and scholars at home and abroad carried out relevant research. Forcellese et al. studied the 3D printing technology of composite materials, formed the composite grid structure with complex shape, and tested its strength [13]. Hao et al. established a fused deposition modeling (FDM) 3D printing platform [14]. On this platform, they used the rotating cylindrical mandrel to make the thermosetting composite a cylindrical grid structure; thus, the use of the mandrel inside the grid structure was saved. These 3D printing technologies can produce more complex

shapes, but it is difficult to produce structures with curved radian shapes and a cylindrical mandrel is still needed as support when making a cylindrical grid structure.

Researchers have also made attempts at innovating the materials of mandrel. Lombardi et al. and Vaidyanathan et al. studied the water-soluble mandrel [15,16]. The mandrel can be dissolved by water, which solves the problem of demolding. However, the water-soluble mandrel cannot be reused. Mohammadi M et al. studied the soft rubber mandrel [17]. Through the grooved rubber mandrel, the prepreg is paved on the groove for curing. Then the volume of the rubber mandrel is reduced by compressed air, so that the rubber core is detached from the cavity to make the cylindrical pyramidal lattice structure. Xie et al. made a rubber mandrel with different thicknesses to make a cap girder and studied the influence of the different rubber mandrels on the thickness error of the cap girder [18]. Li et al. studied the influence of a rubber mandrel with a middle opening and its air pressure on the quality of cap girder through simulation [19]. Although this rubber mandrel can be demolded by its softness, there might be uncontrollable deformation of the external prepreg during curing, finally affecting the accuracy of the formed structure.

Shape memory polymers with lower glass transition temperature (Tg) provide us with a new way of demolding. That is, heating can soften the mandrel so that it can be extracted from the part. Du et al. studied a shape memory polymer mandrel that can be used to form composite parts with complex structures [20,21]. The matrix material of the mandrel is styrene, and the mandrel will gradually soften when heated. It provides a new idea for the manufacture of composites, but its strength cannot support the curing and forming of external prepreg, finally affecting the shape accuracy.

To utilize the specific polymers with heating and softening characteristics and combined materials with sufficient strength that can support modeling, a thermoplastic mandrel composed of carbon fiber-reinforced plastic polymers was made by RTM. The thermoplastic mandrel can support the paving of prepreg at room temperature, complete the curing and molding under high-pressure during high temperatures, and realize heat-softening demolding by its low Tg. The Demolding principle of the thermoplastic mandrel is shown in Figure 1. This paper mainly studies a thermoplastic mandrel for manufacturing composite components with complex structures. Through the manufacturing of complex composite parts including the fabrication of a reusable mandrel and the investigation of the mandrel demolding process, the applicability of a thermoplastic mandrel for manufacturing excellent composite components with complex structure was verified.

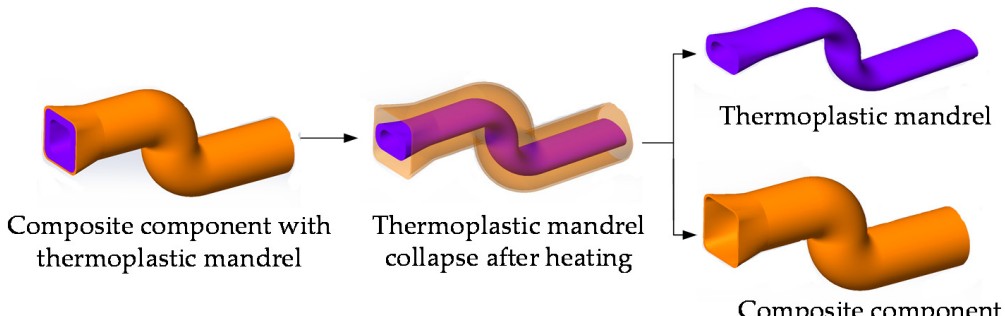

**Figure 1.** Demolding principle of the thermoplastic mandrel.

## 2. Material and Sample Manufacturing

### 2.1. Materials

The mandrel was composed of reinforcing fiber and matrix material. The carbon fiber plain fabric (H3K-CP200, RELIEYSEX, Shanghai, China) was adopted to ensure the strength of the mandrel at room temperature. The matrix material mainly adopted thermoplastic material with low Tg; thus, the mandrel can remain softened without air leakage at high temperature [22–24]. The matrix material consisted of polymethylmethacrylate (PMMA, Elium 188, ARKEMA Co., LTD., Shanghai, China), epoxy resin (NPEF-170, Nan Ya Plastics Corp, Taiwan, China), and liquid nitrile butadiene rubber (LNBR, Lanzhou Petrochemical

Company of Petro china, Shanghai, China). The matrix material was stirred and mixed in the resin reaction kettle at the rate of 60 r/min. Then, the corresponding composite parts were formed by injecting the matrix material into the mold with the paved carbon fiber plain fabric.

Composite parts were planned to be made by prepreg paving and curing. As for the composite material, the carbon fiber/epoxy resin prepreg, VTC401-C650T (VTC401-C650T, Resinas Castro, Pontevedra, Spain) was selected, which is commonly used in the manufacture of composite parts.

### 2.2. Thermoplastic Mandrel Manufacturing

The mandrel was mainly manufactured by RTM, as shown in the Figure 2. Firstly, the metal RTM mold needed to be designed according to the shape of the mandrel. To extract the mandrel, the RTM mold was generally designed in the form of multi-part combination.

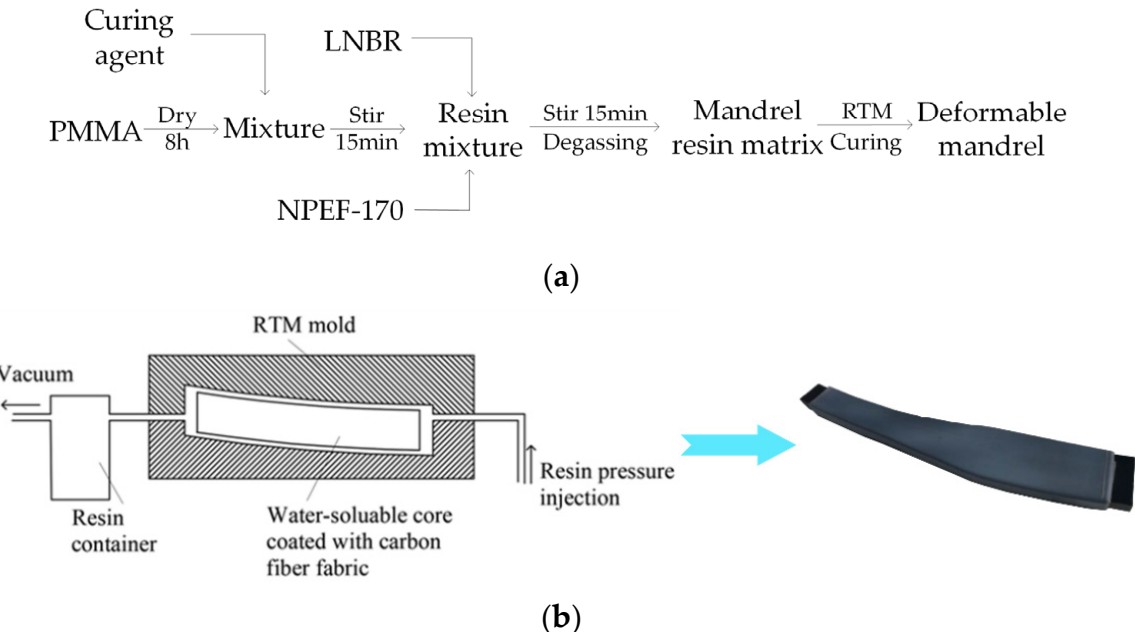

**Figure 2.** RTM process for manufacturing mandrel: (**a**) The flow chart of mandrel preparation; (**b**) Mandrel prepared by RTM process.

Before the RTM process, the matrix material was prepared, referring to the matrix material mentioned in Section 2.1. The materials in the formula were dried in a vacuum oven for 8 h before use. Finally, the resin mixture was centrifuged and deaerated by centrifuge for 10 min, and then it was used to prepare the mandrel by the RTM process. The Mandrel preparation flow chart is shown in Figure 2a.

Because the mandrel often has complex cavities, it is necessary to use a water-soluble core to form the cavity. First of all, we laid twill carbon fiber cloth (TORAY INDUSTRIES, INC., Tokyo, Japan) on the core and then put it into the RTM mold. Next, the matrix material, which was prepared in the previous step (cured at 60 °C for 4 h), was injected into the RTM mold. After the resin was cured, the mandrel was obtained by dissolving the core. The RTM process for manufacturing the mandrel, whose thickness was about 3 mm, is shown in Figure 2b. The gray part on the mandrel is the attached Teflon film. After separating the resin matrix of the mandrel and the prepreg by the film, the prepreg was directly laid on the mandrel.

### 2.3. Manufacturing of Composite Parts

According to the properties of the mandrel, we could use it in the curing process of vacuum bag prepreg to form complex composite components. The end connectors on both

sides of the mandrel were designed to ensure the air tightness of the end connectors being assembled on both sides of the mandrel.

Because the mandrel remained solid at room temperature, after pasting the Teflon layer on the outer surface of the mandrel, the prepreg was directly laid on it and the caul plate (CP) was also laid above the prepreg. A layer of Teflon film was also laid on the inner side of the CP to avoid the bonding between the mandrel and the prepreg. The schematic diagram of the vacuum bag curing process is shown in Figure 3a,b.

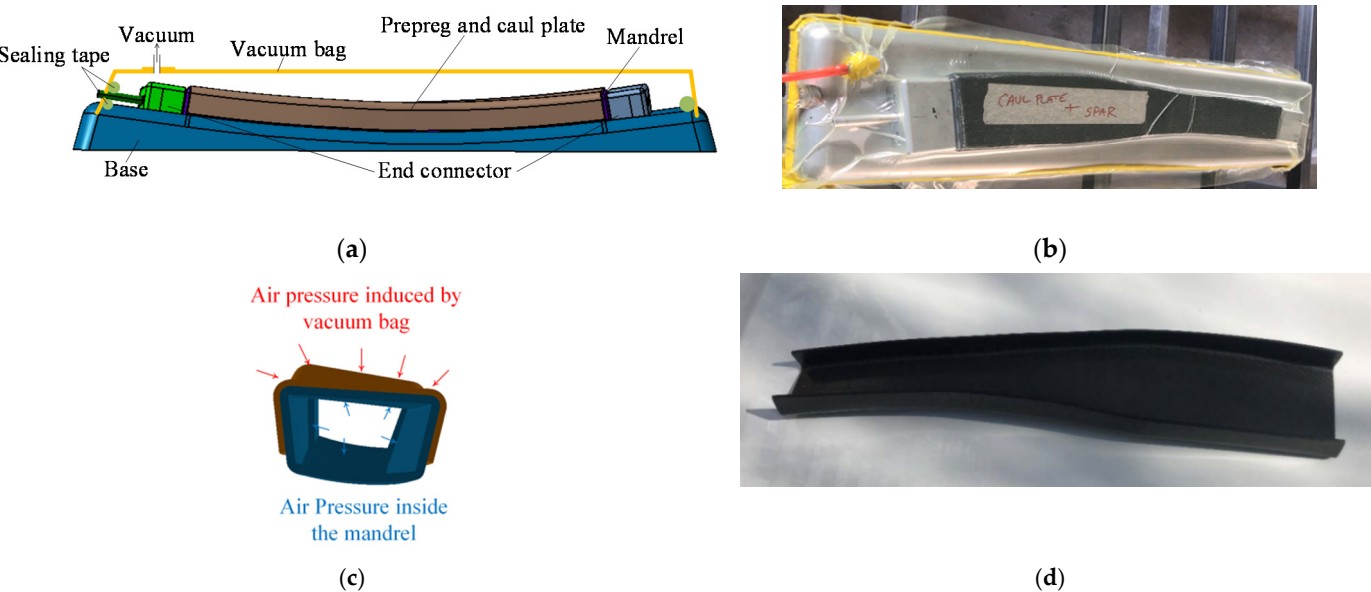

(a)

(b)

(c)

(d)

**Figure 3.** The manufacturing process by vacuum bag technique: (**a**) Schematic diagram of vacuum bag curing process; (**b**) The manufacturing process of composite parts; (**c**) the illustration of air pressure balance during part forming; (**d**) Formed composite parts.

The most important step is to inject air at normal pressure through the ends on both sides of the mandrel, as shown in Figure 3c. If the air pressure inside and outside the vacuum bag is identical to each other, the mandrel can maintain its shape during the curing of the prepreg. After curing the prepreg, the mandrel was heated to soften, and then it was extracted from the cavity part.

As shown in Figure 3d, we used the mandrel for prepreg paving and vacuum bag curing to prepare composite components with complex shapes. The mandrel was sealed by connectors on both sides, and the inflation was maintained in the mandrel's shape when the prepreg was cured.

## 3. Experiments and Tests

Experiments and tests were mainly carried out on the thermoplastic mandrel and the basic properties of composites were prepared with it.

### 3.1. Mandrel Performance Test

The glass transition temperature (Tg), surface roughness, and reusability of thermoplastic mandrel directly determine whether it is suitable for forming composite parts. Therefore, in this section, several experiments were designed to test these three properties.

#### 3.1.1. Glass Transition Temperature

The Tg of the mandrel was tested by dynamic thermomechanical analysis (DMA). The equipment used in the experiment was DMA Q800 (DMA Q800, TA Instruments, New Castle, DE, USA) in tensile mode. The detailed parameters are shown in Table 1 below. Through experiments, we obtained the change curves of the storage modulus and loss modulus of the thermoplastic mandrel with temperature at different heating rates. We then

calculated the peak value of loss modulus and the peak point of derivation of the storage modulus and finally obtained the Tg of thermoplastic mandrel [25,26].

**Table 1.** Dynamic thermodynamics' experimental equipment and conditions.

| Experimental Equipment | Size (mm) | Mode | Control Model | Frequency (Hz) | Heating Rate (°C/min) | Temperature Range (°C) |
|---|---|---|---|---|---|---|
| DMA Q800 (TA Instruments) | 10 × 3.5 × 2.5 | Tensile | Displacement Control | 1 | 1 2.5 5 | 45–140 |

The results of the DMA test of the thermoplastic mandrel are shown in Figure 4. The results showed that the storage modulus decreased more than a thousand times when the mandrel material was heated from 45 to 140 °C. When the material was in low temperature, the resin was at the glass state. After the temperature rose, the material became a highly elastic rubber. In addition, with the increase of the heating rate, the molecular motion needed to overcome large internal friction and thermal motion energy. Therefore, the storage modulus distribution curve moved to the higher temperature side.

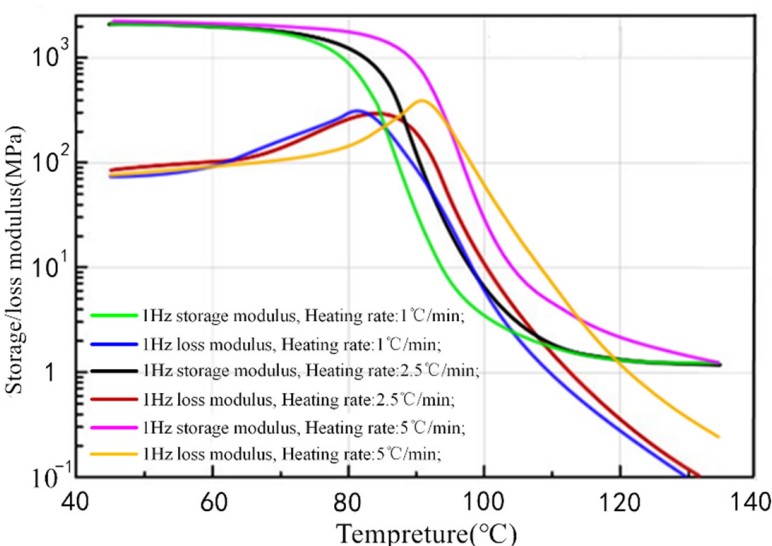

**Figure 4.** DMA test results of mandrel material.

The Tg of the thermoplastic mandrel were determined by the peak value of loss modulus and the peak point of derivation of the storage modulus in the DMA test results [25,26]. At different heating rates, the Tg corresponding to the thermoplastic mandrel is shown in Table 2. The test results showed that the Tg of the mandrel was between 80 and 90 °C, which was 50 to 60 °C lower than that of the thermoset composite. The Tg of the thermoplastic mandrel could avoid the influence of demolding on the formed composite components.

**Table 2.** Glass transition temperature (Tg) of mandrel material.

| Heating Rate | The Peak Value of the Loss Modulus | The Peak Value of Derivative of Storage Modulus |
|---|---|---|
| 1 °C/min | 81 | 81.5 |
| 2.5 °C/min | 86 | 86.5 |
| 5 °C/min | 91 | 90 |

### 3.1.2. Surface Roughness

The surface roughness of the thermoplastic mandrel was mainly tested by a roughness instrument (EDKORS, TR110, CHN). The roughness tester can only measure the roughness within a small length (the sampling length of the instrument is 0.8 mm). Therefore, we tested 20 points on the surface to evaluate its roughness. The points were evenly distributed on the surface of the thermoplastic mandrel, as illustrated in Figure 5.

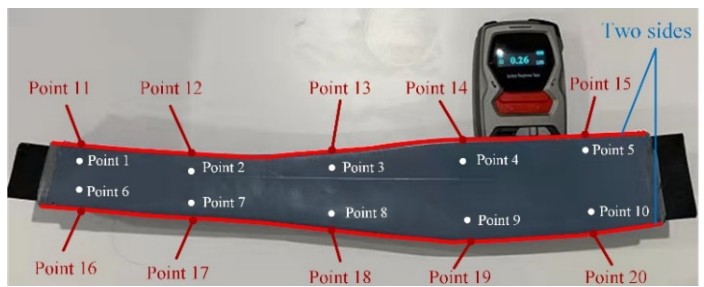

**Figure 5.** Roughness measurement process of each area of mandrel surface.

The surface roughness of the thermoplastic mandrel was tested, and the results of each area are shown in Table 3. It can be seen from the results that, after the mandrel was attached with Teflon film, the roughness of the forming surface was less than Ra 0.5 μm. According to the definition and classification of roughness in the ISO 1302, the surface roughness of the thermoplastic mandrel was N6 and was close to N5. It means that the mandrel had a high degree of surface smoothness.

**Table 3.** Measurement results of mandrel surface roughness.

| Point | 1 | 2 | 3 | 4 | 5 | 6 | 7 | 8 | 9 | 10 |
|---|---|---|---|---|---|---|---|---|---|---|
| Surface Roughness Ra (μm) | 0.35 | 0.2 | 0.27 | 0.26 | 0.37 | 0.17 | 0.25 | 0.29 | 0.28 | 0.27 |
| **Point** | **11** | **12** | **13** | **14** | **15** | **16** | **17** | **18** | **19** | **20** |
| Surface Roughness Ra (μm) | 0.33 | 0.19 | 0.23 | 0.21 | 0.31 | 0.19 | 0.22 | 0.17 | 0.24 | 0.33 |

### 3.1.3. Reusability

In this experiment, we used the reused mandrel to manufacture composite parts and compare the parts' quality to evaluate the reusability of the mandrel. In this study, the mandrel was tested for 20 times by the processes, including softening, deforming, cooling down, hardening, and heating back. As shown in Figure 6 below, the shape after the first, 10th, and 20th deformations is shown.

As shown in the Figure 7, we used the return mold to return the thermoplastic mandrel to the original design shape and manufactured the composite parts again. By means of digital measurement, we measured the profile error of the composite parts manufactured by the mandrel for the first time and composite parts manufactured after 20 times, to determine the reusability of the mandrel.

The equipment used in digital measurement experiments was a Blue Light 3D Scanner (COMET L3D-5M, Steinbichler Optotechnik GmbH, Neubeuern, Germany). After spraying the developer, the complex shape composite components were fixed on the black platform, and the 3D scanner was used to scan the composite components at a certain distance.

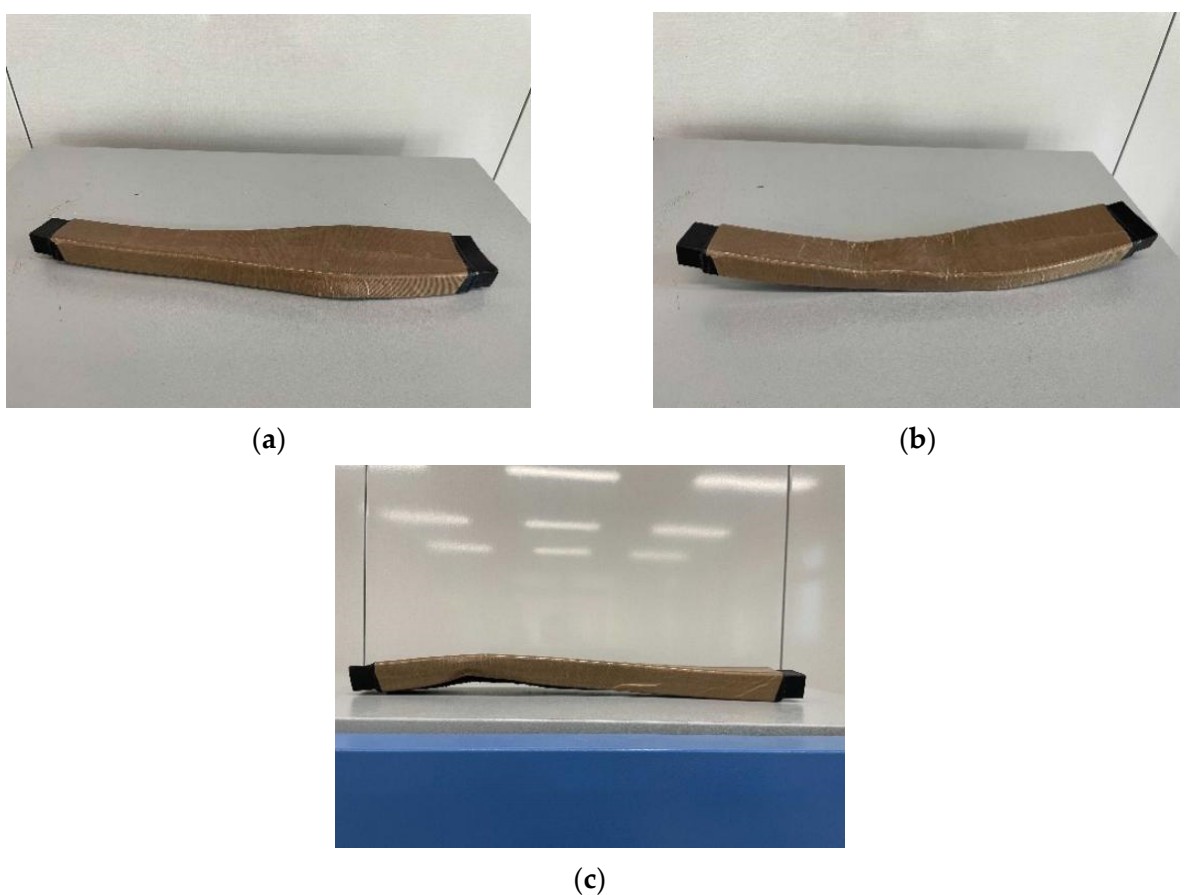

**Figure 6.** Shape of mandrel after deformations: (**a**) Mandrel after the first deformation; (**b**) Mandrel after the 10th deformation; (**c**) Mandrel after the 20th deformation.

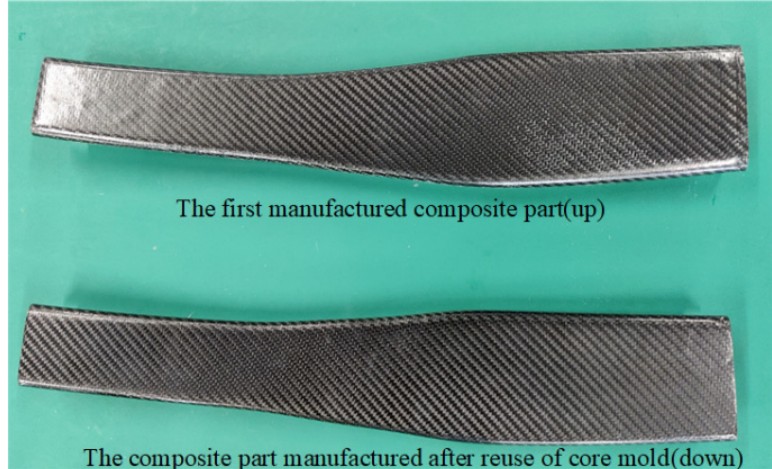

**Figure 7.** Parts formed for the first time by the mandrel and parts prepared after 20 times of mandrel deformation.

Through measurement, we got the point cloud of the physical surface of the part, and then generated the surface. Then, we compared the physical digital model of the component with its design model in Geomagic control software, with the operation of optimal alignment. After the two digital models were aligned, we ran the 3D comparison module to check the error of the surface along the normal direction. Through digital measurement and Geomagic Control comparison, we verified the shape accuracy of the

two parts, and the results are shown in Figure 8. The Figure 8a shows the comparison results of the first composite parts manufactured by the mandrel, and the Figure 8b shows the comparison results of the composite parts prepared after 20 times of mandrel deformation.

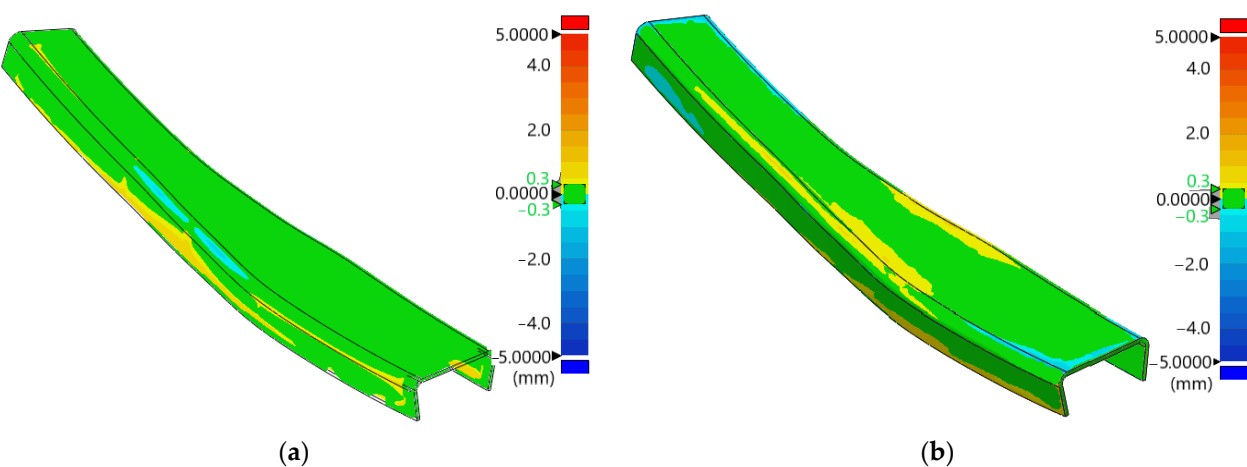

**Figure 8.** Comparison results between composite parts and design digital simulation: (**a**) Composite part first formed by mandrel; (**b**) Composite part prepared after 20 times of mandrel deformation.

It can be seen that after the mandrel was used for 20 times, the internal error of the design digital mold (yellow part) appeared in the middle of the outer surface and the curved parts on both sides. It showed that, after the core mold was reused over 20 times, the mandrel may not be able to return to the design profile. The reason is that after the core mold was reused many times, since the core mold material underwent many transitions between the glass state and the rubber state, a certain residual stress may accumulate in the middle of the outer surface and the curved parts on both sides, which prevents the mandrel from returning to the original design profile when the mandrel is heated. However, the overall molding error can be controlled within 0.3 mm after 20 times, indicating that the mandrel has good reusability and can guarantee the forming accuracy of parts within 20 times.

### 3.2. Performance Test of Composite Parts

In this section, we mainly tested the surface accuracy and internal porosity of the formed composite parts and evaluated the applicability of the thermoplastic mandrel by testing the quality of the parts.

### 3.2.1. Profile Accuracy

The profile accuracy of composite parts was mainly determined by digital measurement experiments, and the specific process of the experiment and the equipment used were introduced in Section 3.1.3. It was compared experimentally with the profile error between the composite part profile formed by the mandrel and the composite part design profile. The visual error distribution diagram and error statistics curve of each point on the part are shown in Figure 9.

It can be seen from Figure 9a that most of the errors of the formed parts were within 0.3 mm, which is marked in green, and part of the error was 0.3–0.5 mm, which is marked with yellow, concentrated on both sides of the "C"-shaped feature of the part. The specific error distribution can be seen from Figure 9b. The average error was 0.04 mm, the maximum error was 0.59 mm, and the points with an error within 0.3 mm accounted for more than 90%. As the error value increased, the probability of the remaining errors gradually decreased and they were nearly symmetrically distributed on both sides of the error mean. The results showed that the shape accuracy of composite parts formed by this thermoplastic

mandrel was high. The mandrel can effectively solve the demolding problem of composite parts and form composite parts with a complex structure.

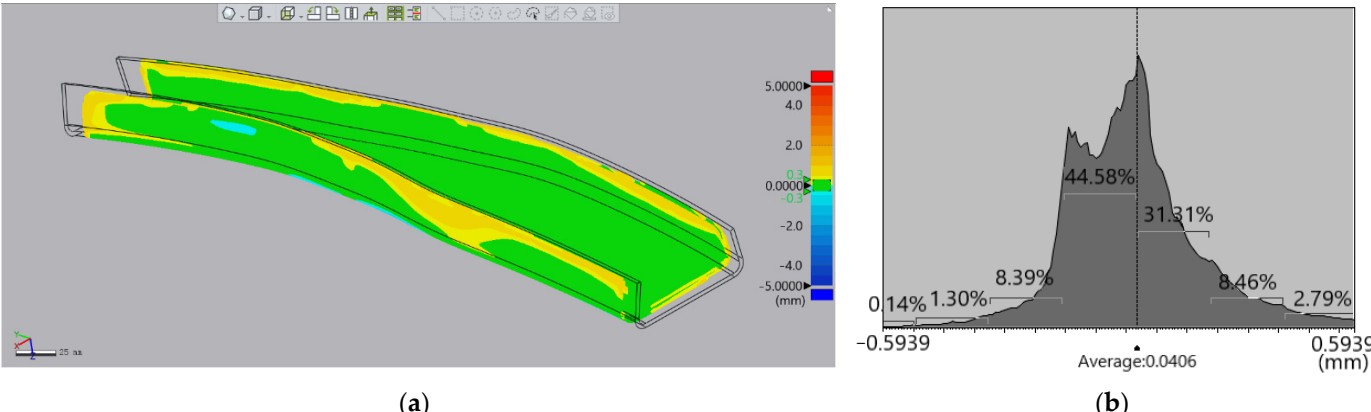

<div style="text-align:center">(<b>a</b>)        (<b>b</b>)</div>

**Figure 9.** Manufacturing error of parts: (**a**) Error distribution diagram of digital analog comparison between parts and design; (**b**) error statistics diagram of digital analog comparison between parts and design.

### 3.2.2. Internal Porosity

The internal porosity of composite parts was tested by industrial CT (RX solutions, EasyTom150Micro, FR). The ray beam passed through the parts, and different results were obtained according to the material distribution inside the parts. According to this result, the CT scanning model of the part was reconstructed, and the defect information such as porosity was obtained. The porosity calculation formula of composite parts is:

$$\text{Porosity} = \text{Defect Volume} / \text{Material Volume} \tag{1}$$

Due to the large volume of the composite parts, they were evenly divided into upper and lower parts for industrial CT measurement. The specific detection parameters are shown in Table 4. Then, the scanning results were processed by the instrument supporting software, and the porosity information of the parts was obtained.

**Table 4.** Industrial CT Detection Parameters.

| Test Parameters | Voltage | Electric Current | Detector Pixel | Projection Number | Voxel | Scan Duration |
|---|---|---|---|---|---|---|
| Figure | 100 kV | 500 μA | 127 μm | 1440 | 95 μm | 56 min |

Figure 10 shows the distribution of internal pores in the upper and lower parts of the composite part measured by industrial CT. It can be seen that the internal pores of the parts were mainly distributed in the middle of the parts. The maximum pore volume in the upper part was 22.46 mm$^3$ and that in the lower part was 4.55 mm$^3$.

The internal porosity of the upper half and the lower half of the composite part is shown in Table 5. The volume of the upper half was 94,798.1093 mm$^3$ and the pore volume was 683.0574 mm$^3$. The volume of the lower half was 82,698.5078 mm$^3$ and the pore volume was 511.0985 mm$^3$. It was calculated that the porosity of the upper half and the lower half of the composite parts was 0.71% and 0.61%.

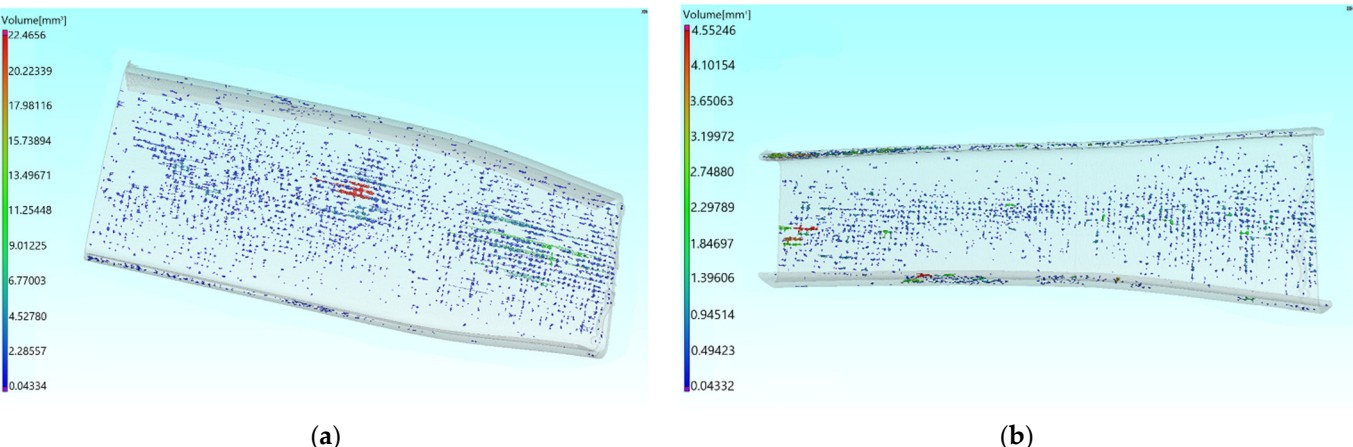

(**a**)                                               (**b**)

**Figure 10.** The distribution of internal pores in the composite part measured by industrial CT: (**a**) The upper parts; (**b**) The lower parts.

**Table 5.** The internal porosity of the upper half and the lower half of the composite part.

| Test Area | Pore Volume (mm$^3$) | Material Volume (mm$^3$) | Internal Porosity |
|---|---|---|---|
| Upper half of part | 683.0574 | 94,798.1093 | 0.72% |
| Lower half of part | 511.0985 | 82,698.5078 | 0.61% |

When we used the mandrel to form parts, the internal and external air pressure of the mandrel was balanced, so the shape of the parts was not extruded. However, through the analysis of the distribution of porosity, it was found that when the part had a free-form surface with large area and small curvature, the compaction effect of thermoplastic mandrel (compared with metal mold) on the prepreg was limited, so there were more fine pores in the middle of the part. Through analyzing the maximum pore volume and porosity, it was seen that the molding internal quality of the lower half was higher than the upper half, but the overall internal porosity of the parts was still far lower than the requirement of 1.5% porosity in the general composite parts' manufacturing standard.

## 4. Conclusions

This paper made a thermoplastic mandrel with heat-softening ability. The mandrel used carbon fiber as the reinforcement material, PMMA and epoxy resin liquid nitrile as the matrix material, and LNBR was added to make the thermoplastic mandrel heat and soften at a certain temperature. Through measurement and experiment on the properties of the thermoplastic mandrel and composite parts manufactured by the mandrel, it was concluded as follows:

- The thermoplastic mandrel made in this paper had suitable glass transition temperature, good surface roughness, and the ability to be reused. According to the experiments of the thermoplastic mandrel, the glass transition temperature of the mandrel was between 80 °C and 90 °C, the surfaces roughness of the mandrel was below Ra 0.5 μm, and the mandrel can be reused more than 20 times, because of the parts manufactured by the mandrel still maintaining high accuracy after the mandrel was reused for 20 times.
- The thermoplastic mandrel can effectively solve the demolding problem and the composite parts manufactured by the mandrel have low profile error and internal porosity. According to the experiments of composite parts, composite parts with complex structure could be manufactured by the mandrel successfully. The average error of the molded part is about 0.04 mm and the average porosity of the upper and lower halves is 0.72% and 0.61%.

The thermoplastic mandrel studied in this paper has suitable Tg, smooth surface roughness, excellent recycling characteristics, and good part-forming performance. Therefore, it can be used for the forming of composite components with complex structure, and makes the application range of carbon fiber composites more extensive.

**Author Contributions:** Methodology, S.C. and X.J.; Field Investigation, S.C., X.J. and F.X.; Data curation, S.C., X.J. and J.A.; Writing—Original Draft Preparation, S.C., C.Z. and J.A.; Writing—Review and Editing, X.J. and C.Z. All authors have read and agreed to the published version of the manuscript.

**Funding:** This research received no external funding.

**Institutional Review Board Statement:** Not applicable.

**Informed Consent Statement:** Not applicable.

**Data Availability Statement:** Data are contained within the article.

**Conflicts of Interest:** The authors have no conflict of interest.

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
