# Peer review of "Thermoplastic Mandrel for Manufacturing Composite Components with Complex Structure"

_aerospace, doi:10.3390/aerospace8120399_

Round 1

Reviewer 1 Report

The paper deals with the preparation of a thermoplastic mandrel with heating softening ability. The proposed paper may attract huge attention.  But, overall the article has some drawbacks I listed below.

  • Use only one bibliographic style according to requirements of the Journal
  • Apply one style for recalling drawings in text “Figure” or “figure”
  • Separate the methodology from the discussion of the results - there will be no mess
  • Line 141 and 142: I would rather place this sentence into Section: Introduction (the end)
  • Section 3.1. seems redundant
  • Line 146: define “Tg” for the readers who are not familiar with this method
  • Table 1: why have you conducted measurements at 3 heating rates? Explain.What differences have you expected?
  • Line 159: you’ve mentioned temperature 25°C. Looking at Figure 4 - the measurements starts at 45°C.
  • Figure 4: a) put these curves on one plot - this would facilitate comparing of results; b) there is no need to write explanation on the chart when the frequency was always 1 Hz
  • Line 169 and Table 2: check these values – from curves (Figure 4) it seems to be different values
  • Line 171: if you mention general engineering add reference
  • Write countries with the names of apparatus
  • Lines 184 and 185: on what basis did you conclude that?
  • Line 199: “The left figure…” delete it- this explanation is placed next to Figure 8
  • Section 3.1.3: Insufficiently described method of measurement and evaluation of results
  • Bypass the first paragraph in Section 3.2.1
  • Line 231: what digital models?
  • Lines 237-245: combine the explanation with Figure 9b and with the values presented there
  • Spacing between Table 4 and line 257
  • Caption under Figure 10: delete “the distribution of internal pores in” at (a) and (b) parts
  • Line 263: put into the methodology
  • Line 265: “The volume” -be more precise, what volume
  • whether accuracy to second decimal place is not an exaggeration (lines 265 and 266 and Table 5)

Author Response

The paper deals with the preparation of a thermoplastic mandrel with heating softening ability. The proposed paper may attract huge attention. But overall, the article has some drawbacks I listed below.

1.(1)Use only one bibliographic style according to requirements of the Journal
(2)Apply one style for recalling drawings in text “Figure” or “figure”
A: Thank you for the suggestion. The new edition has been revised according to the requirements of the journal, applying one style for recalling drawings in text “Figure” or “figure”.
Please check in R1.

2.Separate the methodology from the discussion of the results - there will be no mess.
A: Thank you for the suggestion. In this paper, there are so many experiments involved and the discussion of the results is complex. Therefore, separating the methodology and discussion of the results of each experiment may lead to logical inconsistency. Therefore, I did not make any changes on this point. I hope you can understand and accept my decision.

3.Line 141 and 142: I would rather place this sentence into Section: Introduction (the end)
A: Agree.  As your suggestion, I reconsidered the function of these sentences, thinking that the existence of these sentences is redundant. Therefore, I have deleted them in R1.
Please check Line 140 and 141 in R1.

4.Section 3.1. seems redundant
A: Thank you for the suggestion. In my opinion, the experiments in this paper can be divide into two different kinds. One kind of experiment is the performance test of the thermoplastic mandrel and another kind of experiment is the test of profile accuracy and internal porosity of the composite parts. According to the different experimental objects, the experiments of the paper is divided into section 3.1. and section 3.2. Therefore, I think Section 3.1. and Section 3.2. are not redundant.

5.Line 146: define “Tg” for the readers who are not familiar with this method
A: Agree. According to your suggestion, when "Tg" appears for the first time in the paper, it is defined.
Please check Line 143 in R1.

6.Table 1: why have you conducted measurements at 3 heating rates? Explain. What differences have you expected?
A: Thank you for the comment. The reason for conducting measurements at 3 heating rates is that the mandrel is heated at different heating rates when the thermoplastic mandrel is used to make different composite parts. With the increase of heating rate, the molecular motion needs to overcome large internal friction and thermal motion energy.
Therefore, the storage modulus distribution curve will move to the higher temperature side and the glass transition temperature (Tg) of thermoplastic mandrel will also change. Conducting measurements at 3 heating rates are helpful to determine that the Tg of thermoplastic mandrel is between 50 ℃ to 60 ℃, lower than that of the thermoset composite.

7.(1)Line 159: you’ve mentioned temperature 25°C. Looking at Figure 4 - the measurements start at 45°C.
(2)Figure 4: a) put these curves on one plot - this would facilitate comparing of results; b) there is no need to write explanation on the chart when the frequency was always 1 Hz.
(3)Line 169 and Table 2: check these values – from curves (Figure 4) it seems to be different values.
A: Thank you for the comment and suggestion. Due to my personal negligence, I used the wrong data in the paper. The temperature range of the DMA experimental is 45℃ to 140 ℃.
According to your suggestions I have combined the results and put the curves on one plot.
When the heating rate is 1 ℃ / min, the Tg calculated by the mandrel according to loss module and storage module is 81 ℃ and 81.5 ℃ respectively. When the heating rate is 2.5 ℃ / min, the Tg is 86 ℃ and 86.5 ℃. When the heating rate is 5 ℃ / min, the Tg is 91 ℃ and 90 ℃.
Please check Line 155-170 in R1.

8.Line 171: if you mention general engineering add reference.
A: Thank you for the suggestion. It is not appropriate to be quoted here anymore because of its particularity. Therefore, the sentences corresponding to line 171 in the original paper have been deleted.

9.Write countries with the names of apparatus.
A: Agree. According to your suggestion, the paper explains the country of origin of all apparatus used in the experiment.
Please check Line 148, 175, 197, 253 and 254 in R1.

10.Lines 184 and 185: on what basis did you conclude that?
A: Thank you for the comment. According to your suggestion, I realized that the sentences lines 184 and 185 in the original paper were inaccurate, so the expression here was deleted in the revised paper.
Please check Line 183 in R1.

11.Line 199: “The left figure…” delete it- this explanation is placed next to Figure 8.
A: Thank you for the comment. According to your suggestion, I realized that the sentences lines 184 and 185 in the original paper were inaccurate, so the expression here was deleted in the revised paper.
Please check Line 183 in R1.

12.Section 3.1.3: Insufficiently described method of measurement and evaluation of results.
A: Agree. I made a more systematic and detailed introduction to the reuse process of the thermoplastic mandrels, the methods of digital measurement and the methods of point cloud comparison in Section 3.1.3.
Please check section 3.1.3. in R1.

13.Bypass the first paragraph in Section 3.2.1.
A: Agree. According to your suggestion, the first paragraph in section 3.2.1 in the original paper is adjusted to Section 3.2. as an introduction to the profile accuracy and internal quality of composite parts.
Please check Line 230-232 in R1.

14.Line 231: what digital models?
A: Thank you for the comment. The two digital models described in the original paper refer to the manufacturing model and design model of composite parts obtained by digital measurement. I have revised the paper to make it clearer.
Please check Line 237-238 in R1.

15.ines 237-245: combine the explanation with Figure 9b and with the values presented there.
A: Agree. I make a more in-depth interpretation and analysis of the digital measurement results of composite parts in combination with Figure 9b.
Please check Line 243-254 in R1.

16.Spacing between Table 4 and line 257.
A: Thank you for the suggestion. The format here has been modified.
Please check Line 267 in R1.

17.Caption under Figure 10: delete “the distribution of internal pores in” at (a) and (b) parts.
A: Thank you for the suggestion. The format here has been modified.
Please check Line 271 and 272 in R1.

18.Line 263: put into the methodology.
A: Agree. The calculation method of the internal porosity of the composite material has been put into the methodology
Please check Line 260 and 261 in R1.

19.(1)Line 265: “The volume” -be more precise, what volume.
(2)whether accuracy to second decimal place is not an exaggeration (lines 265 and 266 and Table 5)
A: Agree. I have further refined the data corresponding to lines 265 and 266 and table 5 to four decimal places.
Please check Line 275-277 and table 5 in R1.

Reviewer 2 Report

The authors have developed and evaluated a thermoplastic mandrel for manufacturing complex composite parts. The authors have performed a systematic study by testing the mandrel with different methods and for different properties. The paper is also very well organized and written. The presented work is very useful for the scientific community. I suggest publication of the paper in its present form. The only remark is a request to authors to add a few comments on how their study could be assisted/accelerated by numerical simulation.

Author Response

The authors have developed and evaluated a thermoplastic mandrel for manufacturing complex composite parts. The authors have performed a systematic study by testing the mandrel with different methods and for different properties. The paper is also very well organized and written. The presented work is very useful for the scientific community. I suggest publication of the paper in its present form. The only remark is a request to authors to add a few comments on how their study could be assisted/accelerated by numerical simulation.
A: Thank you for the suggestion. The numerical simulation of composite forming process has always been the focus of our research. At present, the material constitutive equation, thermochemical model and stress-strain model have been developed for thermosetting composites.
However, the characteristics of thermoplastic materials are slightly different from those of thermosetting materials, it is necessary to develop the corresponding material constitutive model to quickly and accurately describe the performance changes in the forming of thermoplastic mandrel.
In the future, the research will focus on the molding process of thermoplastic composite mandrel, and develop an efficient and accurate numerical simulation model to help improve the efficiency of related research.

This manuscript is a resubmission of an earlier submission. The following is a list of the peer review reports and author responses from that submission.

Round 1

Reviewer 1 Report

The reviewed article presents the results of research on a new mandrel used in the production of composite elements.  The subject of the article fits into the scope of the journal. The presented research is interesting, but the article requires some corrections and additions before it is accepted for publication.
 1. There is no given conditions under which the mandrel matrix was prepared.
 2. Information on the methodology of research should be collected in one place in the "methodology" section.
 3. Line 171-180.  How a change in the heating rate in the DMA would have an effect on
 internal friction in the material?  I believe that the observed changes in parameters are rather due to the phenomenon of "thermal lag", which is typical for the thermal analysis of polymeric materials. The thermal log affects Tg in the same way. Why were different heating rates used in the DMA tests?  I don't see any point in duch action.
 4. Much attention has been paid to research of roughness. Why is it so important?  In addition to roughness at specific points, it would also be useful to specify the average roughness of the entire surface.  Is it better if the mandrel have large or small roughness?
 5. Table 4 is redundant.  Overall there are too many figures in reviewed work.  Some (4, 8, 10, 15) do not add anything significant.
 6. Line 308-310.  Information about mechanical properties should be included in the main body of the article.  On what basis, however, the authors draw conclusions about excellent properties mechanical?  DMA tests were performed and described in terms of thermal properties.

Reviewer 2 Report

The scientific paper is good for publishing. In figure 9 the results appear to exceed the 120 degree. The dynamic thermomechanical analysis method is not completely described. The text should be verified for spelling errors.

Reviewer 3 Report

see attached file
